# Adherence to Clinical Practice Guidelines and Colorectal Cancer Survival: A Retrospective High-Resolution Population-Based Study in Spain

**DOI:** 10.3390/ijerph17186697

**Published:** 2020-09-14

**Authors:** Francisco Carrasco-Peña, Eloisa Bayo-Lozano, Miguel Rodríguez-Barranco, Dafina Petrova, Rafael Marcos-Gragera, Maria Carmen Carmona-Garcia, Josep Maria Borras, Maria-José Sánchez

**Affiliations:** 1Radiation Oncology Department, University Hospital Virgen Macarena, 41009 Sevilla, Spain; francisco_cp84@hotmail.com (F.C.-P.); eloisabayo@gmail.com (E.B.-L.); 2Escuela Andaluza de Salud Pública, 18011 Granada, Spain; miguel.rodriguez.barranco.easp@juntadeandalucia.es (M.R.-B.); mariajose.sanchez.easp@juntadeandalucia.es (M.-J.S.); 3Instituto de Investigación Biosanitaria ibs.GRANADA, 18012 Granada, Spain; 4CIBER de Epidemiologia y Salud Pública (CIBERESP), 28029 Madrid, Spain; rafael.marcos@udg.edu; 5Medical Sciences Department, Faculty of Medicine, University of Girona (UdG), 17071 Girona, Spain; 6Epidemiology Unit and Girona Cancer Registry, Oncology Coordination Plan, Catalan Institute of Oncology, Department of Health, Government of Catalonia, 17007 Girona, Spain; 7Descriptive Epidemiology, Genetics and Cancer Prevention Group, Girona Biomedical Research Institute—IDIBGI, Salt, 17190 Girona, Spain; mccarmonagarcia22@gmail.com; 8Medical Oncology Department, Catalan Institute of Oncology, University Hospital Dr Josep Trueta, 17007 Girona, Spain; 9Department of Clinical Sciences, IDIBELL, University of Barcelona, Hospitalet, 08908 Barcelona, Spain; jmborras@idibell.cat; 10Department of Health, Catalonian Cancer Strategy, Hospitalet, 08908 Barcelona, Spain; 11Department of Preventive Medicine and Public Health, University of Granada, 18010 Granada, Spain

**Keywords:** colorectal cancer, adherence, clinical practice guidelines, population-based study, high-resolution study, cancer survival

## Abstract

Colorectal cancer (CRC) is the third most common cancer worldwide. Population-based, high-resolution studies are essential for the continuous evaluation and updating of diagnosis and treatment standards. This study aimed to assess adherence to clinical practice guidelines and investigate its relationship with survival. We conducted a retrospective high-resolution population-based study of 1050 incident CRC cases from the cancer registries of Granada and Girona, with a 5-year follow-up. We recorded clinical, diagnostic, and treatment-related information and assessed adherence to nine quality indicators of the relevant CRC guidelines. Overall adherence (on at least 75% of the indicators) significantly reduced the excess risk of death (RER) = 0.35 [95% confidence interval (CI) 0.28–0.45]. Analysis of the separate indicators showed that patients for whom complementary imaging tests were requested had better survival, RER = 0.58 [95% CI 0.46–0.73], as did patients with stage III colon cancer who underwent adjuvant chemotherapy, RER = 0.33, [95% CI 0.16–0.70]. Adherence to clinical practice guidelines can reduce the excess risk of dying from CRC by 65% [95% CI 55–72%]. Ordering complementary imagining tests that improve staging and treatment choice for all CRC patients and adjuvant chemotherapy for stage III colon cancer patients could be especially important. In contrast, controlled delays in starting some treatments appear not to decrease survival.

## 1. Introduction

Colorectal cancer (CRC) is the third most incident cancer worldwide in both sexes responsible for 10.2% of all cancer cases, after lung cancer (11.6%) and breast cancer (11.6%) [1]. According to the Spanish Network of Cancer Registries (REDECAN), in 2020 in Spain there will be 44,231 new CRC cases [2]. CRC will be the most frequent cancer considering both sexes and the second most frequent tumor after prostate cancer in men and breast cancer in women [2].

In relation to other European countries, in Spain CRC occupies an intermediate position in mortality. In particular, in 2018 15,167 people died from CRC, which was the second leading cause of death due to cancer in both sexes, representing 13.5% of cancer deaths in men and 13.7% of cancer deaths in women (www.mscbs.gob.es). In the European cancer registry-based study on survival and care of cancer patients (EUROCARE), for the period 2000–2007, CRC in Spain had an observed and relative 5-year-survival of 45.66% and 55.32%, respectively, in men and 47.62% and 55.01%, respectively, in women [3]. 

Research shows that two main factors influence CRC survival: the stage of the disease at the time of diagnosis and adherence to the relevant clinical practice guidelines regarding diagnosis and treatment. For instance, a key study by Allemani et al. [4] investigated why population-based CRC survival in the late 1990s was better in the United States compared to Europe, and concluded that the stage at diagnosis and adherence to clinical practice guidelines were the main causes of these differences. Similar conclusions were reached by Gatta et al. [5] in a study based on eleven European Cancer Registries.

Clinical practice guidelines define the recommended actions at each moment of the healthcare process based on the best scientific evidence available and thus reduce unwarranted variability in diagnostic testing and treatment. Population-based high-resolution studies, in which cancer registries systematically collect detailed clinical and pathological data beyond what is routinely recorded, are one of the best tools to examine how adherence to guidelines influences survival. In contrast to clinical studies which frequently exclude patients with advanced age, comorbidities, or lower socio-economic status, population-based studies include all patients in a given jurisdiction and are less prone to selection and referral biases [6]. This makes them an essential resource for the continuous evaluation and updating of diagnosis and treatment standards. 

The aim of this study was to analyze the degree of adherence to clinical practice guidelines for CRC and investigate its relationship with survival in all incident CRC cases diagnosed in 2011 in two provinces in Spain. To our best knowledge, this is the first high-resolution population-based study to examine in detail adherence to CRC clinical practice guidelines in Spain.

## 2. Materials and Methods

We conducted a retrospective high-resolution population-based cohort study of all CRC cases (C18–C20 according to the 3rd edition of the International Classification of disease for Oncology, CIE-O-3 [7]), diagnosed during 2011 in persons older than 15 and residing in the provinces of Granada and Girona (Spain). The lower age limit was set at 15 to avoid distortions produced by the generally very low mortality in younger age groups. Granada and Girona were selected among the seven Spanish cancer registries participating in the European High-Resolution studies to represent Southern (Granada) and Northern (Girona) Spain and because they have similar population sizes, thereby contributing a similar number of CRC cases to the analysis.

### 2.1. Information Sources

The information was obtained from the Cancer Registries of Granada and Girona, which are both accredited by the International Agency for Research on Cancer (IARC). The registries record all cases of invasive cancer diagnosed for the first time in residents of the provinces of Granada and Girona, each with a population of about 920,000 and 760,000 inhabitants, respectively.

The information in both registries comes from public and private health centers in both provinces. The detection of cancer cases is based mostly on the information that comes from the Basic Minimum Data Set of hospital discharges (Conjunto Mínimo Básico de Datos, CMBD), and the Pathological Services. Other sources of information are the medical records provided by other hospital services in which cancer patients are diagnosed and/or treated.

### 2.2. Variables

Following the specific European High-Resolution Studies (http://www.hrstudies.eu/) protocols, trained cancer registries personnel accessed the clinical records of each case to confirm the CRC diagnosis and record sociodemographic, tumor characteristics, diagnostic, and treatment related data. The following variables were used:

Demographic and clinical variables. Sex, age at the time of diagnosis; date of incidence according to recommendations of the European Network of Cancer Registries (ENCR); the most valid method of diagnosis coded according to the ENCR (clinical, microscopic confirmation, imaging tests, biomarkers, autopsy); diagnostic modality (symptomatic, by screening); topography (anatomical site and subsite); tumor morphology (coded according to the International Classification of Diseases for Oncology (ICD-O)—3rd edition); degree of differentiation (I-well differentiated; II-moderately differentiated; III-poorly differentiated; IV-undifferentiated); stage at the time of diagnosis (clinical and post-surgical TNM, based on TNM Classification, 7th edition [8]); number of lymph nodes affected and number of lymph nodes examined; comorbidities based on the Charlson index [9]; and tobacco use (current smoker, ex-smoker, non-smoker).

Diagnostic examinations performed. We recorded whether the patients had undergone each of the following procedures: colonoscopy, barium enema, computed tomography, colonographic magnetic resonance imaging (MRI), extension preoperative study (liver, lung, brain, and bone with ultrasound, chest radiography, nuclear magnetic resonance (NMR), chest computed tomography (CT), echoendoscopy, abdominal CT).

Treatment-related variables. We recorded whether patients had undergone each of the following treatments (with type and mode): surgery, chemotherapy, radiotherapy, and targeted therapy.

Adherence to clinical practice guidelines. The relevant guidelines for both provinces were the Integrated Healthcare Process for Colorectal Cancer published by the Health Agency of Andalucía in 2011 [10] (for Granada) and the Onco-Guide of Colon and Rectum published in 2008 by the Health Department of the Catalan Government [11] (for Girona). Both documents define recommendations related to the diagnosis, treatment, and care of people diagnosed with CRC. In addition, both documents are based on international guidelines and offer similar recommendations. The quality indicators (QIs) used to measure adherence were those for which information was included in the European High-Resolution Studies protocols. 

The specific QIs were the following: (QI1) whether the patient, following the CRC diagnosis, was assessed by the specific tumor commission before starting treatment; (QI2) whether the following complementary imaging tests were requested: colonoscopy, CT of the chest, abdomen and pelvis, and MRI of the pelvis; (QI3) whether the patient underwent surgery sooner than 30 days after the histological diagnosis; (QI4) whether the patient initiated neoadjuvant therapy (radiotherapy/chemotherapy or chemotherapy or radiotherapy) sooner than 30 days after the histological diagnosis; (QI5) whether the patient started adjuvant treatment sooner than 8 weeks after surgical treatment; (QI6) whether the patient underwent excision and analysis of at least 12 lymph nodes to allow for appropriate lymph node staging; (QI7) whether the patient, if diagnosed with a stage III colon carcinoma, underwent chemotherapy treatment; (QI8) whether the patient, if diagnosed with stage II or III carcinoma of the rectum underwent radiotherapy/chemotherapy or radiotherapy treatment with neoadjuvant or adjuvant intent; and (QI9) whether perioperative mortality occurred, defined as patient death in the first 30 days after surgical treatment.

We calculated adherence for each indicator (QI1 to QI9) and overall adherence which was defined as adherence on at least 75% of the indicators that apply to each patient. 

Vital status. Patient follow-up was updated until 31 December 2016, based on the National Death Registry and the patients’ medical records, whereby cases reported only in the death certificate or identified by autopsy were excluded.

### 2.3. Statistical Analyses

We first describe the demographic and clinical characteristics of the sample using absolute and relative frequencies, also stratified as a function of tumor subsite: colon cancer (CC) or rectal cancer (RC). Significant differences in characteristics between subsites were contrasted by means of the Chi-Square test or the Fisher’s exact test (according to the fulfillment of the application conditions).

To analyze the relationship between adherence and survival, we calculated observed and net survival at 1, 3, and 5 years since the diagnosis of colorectal cancer and computed the relative excess risk of death (RER) as a function of adherence to the quality indicators. In particular, observed survival was calculated using the Kaplan–Meier method. Regarding net survival, to eliminate the possibility of death from other causes, this was calculated using the Pohar-Perme estimator [12], which represents the hypothetical survival that patients would have had if their cancer had been the only possible cause of death. For the calculation of the net survival, we used life tables and general mortality using the Elandt-Johnson method [13]. To estimate the RERs, we used generalized linear models with a Poisson error structure based on collapsed data using exact survival times in the net survival framework. RERs with 95% confidence intervals (CIs) were estimated by the studied factors using the maximum likelihood method. 

Finally, we also performed analyses restricted to patients diagnosed with stage II or III disease because of the potentially stronger impact of the clinical guidelines in this group.

All statistical analyses were conducted in Stata v14 (StataCorp LP. 2015, College Station, TX, USA).

## 3. Results

A total of 1050 patients diagnosed with CRC (33.6% with RC and 66.4% with colon cancer CC) were included in the study. Table 1 shows patients’ characteristics at diagnosis, also as a function of tumor location. The majority of patients (60.9%) were men and aged above 65 (67.0%). Median age at diagnosis was 71 (interquartile range (IR): 54–88): 71 in men (IR: 54–88) and 72 in women (IR: 52–92). More than half of the patients (51.5%) presented at a late stage (TNM III or IV) (see Table 1). 

The diagnostic exams and preoperative tests performed are listed in Table 2. Table 3 shows the treatments undergone for all CRC cases and as a function of location (CC or RC) and Appendix A show the combination of treatments administered.

### 3.1. Adherence to Clinical Practice Guidelines for Colorectal Cancer (CRC)

The adherence was calculated after excluding cases with missing data on each quality indicator. Overall adherence (defined as adherence on ≥75% of the indicators) was observed for 74.7% of the patients; for 19.5% of patients there was adherence on 50–74% of the indicators, and for 5.8% on ≤49% of the indicators.

The results for the separate indicators are shown in Table 4. Those with highest adherence were QI1, QI9, and QI6. In particular, 91.9% of CRC patients were evaluated by the specific tumor commission before staring treatment (QI1), 94.2% did not suffer perioperative mortality (QI9), and 74.6% had at least 12 lymph nodes excised and analyzed (QI6).

In contrast, the indicator with the worst adherence was QI4. In particular, only 34.9% of CRC patients started neoadjuvant therapy (radiotherapy/chemotherapy or chemotherapy or radiotherapy) sooner than 30 days after histological diagnosis (QI4). The mean and median number of days elapsed until neoadjuvant treatment start were 47 and 43, respectively (39 and 41 for CC and 48 and 44 for RC).

The rest of the indicators showed that for 62.0% of CRC patients the recommended complementary imaging tests were requested (QI2). Sixty-two percent underwent surgery sooner than 30 days after the histological diagnosis (QI3), with mean and median number of days elapsed until surgery of 36 and 26, respectively (33 and 20 for CC, and 45 and 37 for RC). Sixty-four percent of CRC patients started adjuvant treatment sooner than 8 weeks after surgical treatment (QI5). The mean and median number of days elapsed until adjuvant treatment start were 51 and 45, respectively (49 and 43 for CC and 59 and 56 for RC). Finally, 66.7% of patients diagnosed with a stage III CC underwent chemotherapy (QI7) and the percentage of RC patients with a diagnosis of stage II and III who underwent radiotherapy/chemotherapy or radiotherapy treatment with neoadjuvant or adjuvant intent was 73.4% (QI8).

### 3.2. Survival Analysis

Appendix A show the observed and net survival of patients 1, 3, and 5 years after diagnosis as a function of demographic, clinical, and tumor characteristics. Stage at diagnosis was a strong determinant of survival (*p*-values ≤ 0.020). In particular, CRC patients diagnosed with Stage I disease had net survival of 95% at 1, 92% at 3, and 90% at 5 years after diagnosis. In contrast, patients diagnosed with stage IV disease had a net survival of only 50% at 1, 22% at 3, and 11% at 5 years after diagnosis (*p* < 0.001 for Stage I vs. Stage IV comparison). The same pattern was observed both for CC and RC cases separately (see Appendix A, respectively). Survival was also poorer in men compared to women (*p* = 0.015), in older compared to younger patients (*p* < 0.001 for “75+” vs. “<65”, and *p* = 0.012 for “65–74” vs. “<65” groups), and in patients with a higher comorbidity burden (*p* < 0.001 for high vs. no comorbidity and *p* = 0.034 for low vs. no comorbidity) (see Appendix A).

Table 5 shows observed and net survival as a function of adherence to the QIs. Overall adherence (on ≥75% of indicators) reduced the excess risk of death by 65% (RER = 0.35, 95% CI 0.28–0.45, *p* < 0.001). This result was confirmed in an analysis restricted only to patients diagnosed with stage II or III disease (RER = 0.41, 95% CI 0.25–0.67. *p* < 0.001), suggesting almost 60% reduced excess risk of dying for cases where guidelines were followed (see Appendix A). Considering the individual indicators, significant differences were observed on QI2, QI3, and QI7. In particular, patients for whom the recommended complementary imaging tests were requested (QI2) (RER = 0.58, 95% CI 0.46–0.73, *p* < 0.001) and patients diagnosed with a stage III CC who underwent chemotherapy (QI7) (RER = 0.33, 95% CI 0.16–0.70, *p* = 0.004) had a reduced excess risk of death. Patients who underwent surgery sooner than 30 days after the histological diagnosis had an increased excess risk of death (RER = 1.77, 95% CI 1.17–2.66, *p* = 0.007) but this difference was not significant when analysis was restricted to patients diagnosed in stages II and III (RER = 2.30, 95% CI 0.99–5.32, *p* = 0.053).

## 4. Discussion

This population-based study of patients diagnosed with primary invasive CRC during 2011 in two provinces (Granada and Girona) in Spain analyzed the adherence to clinical practice guidelines regarding diagnosis and treatment and the effect of adherence on survival up to five years after diagnosis. Results showed that overall adherence to the clinical practice guidelines (on ≥75% of indicators) improved survival, reducing excess risk of death by 60–65% (depending on whether all patients are considered or only those diagnosed in stages II and III, respectively). Detailed analyses of the separate indicators suggested that ordering complementary imagining tests that improve staging and treatment choice for all CRC patients and adjuvant chemotherapy for stage III colon cancer patients could improve survival. In contrast, controlled delays in starting some treatments appeared not to decrease survival.

Adherence to clinical practice guidelines for CRC has recently been examined in other countries including Canada [14] and The Netherlands [15]. In particular, a population-based study in a Canadian province examined adherence to adjuvant chemotherapy in patients with stage II or III colorectal cancer [14], whereas in The Netherlands a survey of medical oncologists examined adherence to clinical guidelines for systemic treatment for high-risk stage II and III colon and metastatic colorectal cancer [15]. However, these studies did not investigate the relationship between adherence and survival. Hence, the current study adds valuable information regarding the implications of a broad set of clinical guidelines for patient survival, using population-based data and including all patients in the selected regions, regardless of stage at diagnosis.

In the current study, more than half of patients were diagnosed at later stages (26.4% stage III and 25.1% stage IV). The observed distribution by stage was similar to that recorded by Minicozzi et al. [16] in an analysis of the differences in the stage and treatment of CRC in Italy and France, and in other publications [4,17,18]. Stage at diagnosis was strongly related to survival, in line with previous results by Gatta et al. [5], confirming stage at diagnosis as one of the strongest determinants of survival.

The analysis of adherence to clinical guidelines on the separate indicators showed significant improvement in survival only on two indicators (QI2 and QI7). In particular, considering the individual indicators, QI1 (assessment by the specific tumor commission before starting treatment) was not associated with survival, which could be due to the high adherence in the current study (91.9%), which is also higher than that reported in other studies (e.g., 70.1% in Munro et al. [19]).

Complementary imaging studies (colonoscopy, CT of the chest, abdomen and pelvis, and MRI of the pelvis) were requested for 62% of patients (QI2). These patients had better survival, in particular a 42% reduced excess risk of dying, compared to patients for whom no complementary imaging studies were requested. This result could be explained by the additional information provided by the complementary tests, which could have improved the staging of the lesion and helped improve the choice of treatment. This is supported by studies examining the role of CT of the chest, abdomen, and pelvis as part of an extended diagnostic examination or in studies evaluating the importance of pelvic MRI or colonoscopy in local staging, with repercussions for survival [20].

Regarding QI3, net survival was higher in the group of patients in whom the surgery was performed later than the established limit. This could be due to the proximity of the mean and median values observed in the sample (36 and 26, respectively) to the date established as the limit by the guidelines (30 days). Our study and previous findings corroborate that a controlled delay of the surgical treatment does not have an impact on survival [21]. The lack of effect of QI4 (starting neoadjuvant treatment sooner than 30 days after the histological diagnosis) suggests that also a controlled delay in starting neoadjuvant therapy may have no impact on survival.

Multiple studies have analyzed the delay in administering the first treatment and its impact on the risk of death from CRC without reaching definitive conclusions. While some studies establish a clear relationship between survival and delay greater than 30 days [22], others do not find a difference [23] and others find no differences in survival for a period of up to 34 weeks [24].

The percentage of patients who started adjuvant treatment sooner than 8 weeks after surgical treatment was 63.9% (QI5). The cut-off considered in the guidelines was 6 weeks, however, different studies and meta-analyses [25] have established 8 weeks as the time limit after which a delay in the start of chemotherapy would have an impact on survival, so we used 8 weeks as a criterion. However, in our study there were no significant differences in survival when the cut-off point was set at 6 or 8 weeks, nor when the analysis was restricted to patients in stages II and III when the benefit of chemotherapy is clearly established. Again, a possible explanation for the lack of significant differences may be the proximity of the mean and median waiting times in our study (51 and 45 days, respectively) to the cut-off established by the guidelines (56 days or 8 weeks).

Scientific evidence establishes that at least 12 lymph nodes must be sampled to perform adequate staging, considering it an independent prognostic factor and key in decision-making for patients who can benefit from adjuvant chemotherapy, especially those diagnosed with stage II and III CC [26]. This criterion was met for 74.6% of patients in the current study (QI6), who also had somewhat better survival. However, as was the case in other studies such as those by Berberoglu et al. [27], the difference was not significant.

QI7 showed that 66.7% of CC patients diagnosed with stage III disease underwent chemotherapy and this was associated with better survival in all scenarios analyzed, whereby the excess risk of death was reduced by 67%. Chemotherapy improves local control, disease-free survival [28], and global survival [29]. This is why systemic treatment should be a standard for this group of patients. However, in our study only 66.7% of patients underwent adjuvant treatment, a percentage that is still greater than that reported by other European [4] and North-American [14] cancer registries.

The relevant guidelines for our study population established that patients diagnosed with stage II or III RC should undergo radiotherapy/chemotherapy or radiotherapy treatment with neoadjuvant or adjuvant intent. The adherence to this recommendation was assessed by IQ8 at 73.4%, and patients who adhered to this indicator showed better survival. However, contrary to results published by Peng et al. [30], this difference was not significant.

The percentage of patients who died in the first 30 days after surgery was 5.8% (QI9), a result similar to that found in The Netherlands [31]. It is still above the average published by the Spanish Association of Surgeons, which is 1.8% [32]; however, their estimate was not based on population-based data and was not externally audited. Excluding urgent surgery (20% of cases) from the estimate resulted in 2.9% perioperative mortality (3.3% for CC and 2.3% for RC).

Overall, results published by other US and European registries regarding perioperative mortality vary between 2% and 6% [4]. Some of the factors that have an important influence on perioperative mortality are age and comorbidities as can be evidenced by the study of Chin-Chia et al. [33]. In this study patients who were older and had a higher Charlson comorbidity index had 106% higher risk of death, independent of sex, socioeconomic status, demographic region, and treatment modality (neoadjuvant or adjuvant). In our study, perioperative mortality was higher among older patients, patients with a higher Charlson comorbidities index, and patients who underwent neoadjuvant (vs. adjuvant) treatment (see Appendix A).

A limitation of the current study was that it was based on data from two provinces only. However, this was the first population-based high-resolution study examining adherence to clinical practice guidelines for CRC and survival in Spain. In addition, comparing the results across provinces was not among the goals of the current study but it should be addressed in future research because regional differences in survival have been observed for other cancers [34]. It would be especially relevant to do this for the regions with highest incidence and mortality from colorectal cancer and further benchmark it with results from other countries. There is important variability in treatments administered both within and between countries, as is the case for radiotherapy [35,36,37] and chemotherapy [15].

## 5. Conclusions

Survival was strongly influenced by the stage at diagnosis and adherence to the clinical guidelines. The specific guidelines that showed significant differences were ordering complementary imaging tests and undergoing adjuvant chemotherapy treatment in the case of patients diagnosed with a stage III CC. In contrast, controlled delays in starting some treatments appeared not to decrease survival. Nevertheless, overall adherence (on ≥75% of the indicators) also showed improved CRC survival.

## Figures and Tables

**Table 1 ijerph-17-06697-t001:** Distribution of colorectal cancer patients according to characteristics at diagnosis.

		Sub-Site						
		Colorectal		Colon		Rectum		*p*-Value
		*n*	(%)	*n*	(%)	*n*	(%)	
**Total**		1050	(100.0)	697	(100.0)	353	(100.0)	
Gender	Male	639	(60.9)	425	(61.0)	214	(60.6)	
	Female	411	(39.1)	272	(39.0)	139	(39.4)	0.912
Age group	<65	346	(33.0)	223	(32.0)	123	(34.8)	
	65–74	272	(25.9)	182	(26.1)	90	(25.5)	
	75+	432	(41.1)	292	(41.9)	140	(39.7)	0.678
Charlson comorbidity index	No comorbidity (0–1)	517	(49.2)	328	(47.1)	189	(53.5)	
	Low comorbidity (2)	165	(15.7)	113	(16.2)	52	(14.7)	
	High comorbidity (3+)	368	(35.0)	256	(36.7)	112	(31.7)	0.135
Smoker	Yes, currently	129	(13.9)	76	(12.3)	53	(17.3)	
	Yes, previously	297	(32.1)	199	(32.1)	98	(31.9)	
	No, never	500	(54.0)	344	(55.6)	156	(50.8)	0.104
Grading	Grade I, well differentiated	165	(15.7)	105	(15.1)	60	(17.0)	
	Grade II, moderately differentiated	595	(56.7)	400	(57.4)	195	(55.2)	
	Grade III, poorly differentiated	90	(8.6)	68	(9.8)	22	(6.2)	
	Grade IV, undifferentiated	6	(0.6)	4	(0.6)	2	(0.6)	
	Not determined, not graded	194	(18.5)	120	(17.2)	74	(21.0)	0.187
Modality of diagnosis	Symptomatic tumour	1030	(98.3)	683	(98.1)	347	(98.6)	
	Screened-detected	18	(1.7)	13	(1.9)	5	(1.4)	0.599
Multifocality	Yes	42	(4.0)	35	(5.1)	7	(2.0)	
	No	1001	(96.0)	658	(94.9)	343	(98.0)	0.018
Basis of diagnosis	DCO	1	(0.1)	0	(0.0)	1	(0.3)	
	Clinical	39	(3.7)	31	(4.4)	8	(2.3)	
	Microscopic	1010	(96.2)	666	(95.6)	344	(97.5)	0.054
Histological type	Adenocarcinoma	974	(92.8)	648	(93.0)	326	(92.4)	
	Other	76	(7.2)	49	(7.0)	27	(7.6)	0.715
TNM7 Stage	I	179	(17.0)	111	(15.9)	68	(19.3)	
	II	278	(26.5)	203	(29.1)	75	(21.2)	
	III	277	(26.4)	166	(23.8)	111	(31.4)	
	IV	264	(25.1)	183	(26.3)	81	(22.9)	
	Unknown	52	(5.0)	34	(4.9)	18	(5.1)	0.010
T	T1	107	(10.2)	73	(10.5)	34	(9.6)	
	T2	98	(9.3)	60	(8.6)	38	(10.8)	
	T3	572	(54.5)	369	(52.9)	203	(57.5)	
	T4	183	(17.4)	126	(18.1)	57	(16.1)	
	Tx	90	(8.6)	69	(9.9)	21	(5.9)	0.136
N	N0	482	(45.9)	349	(50.1)	133	(37.7)	
	N1	229	(21.8)	150	(21.5)	79	(22.4)	
	N2/N+	213	(20.3)	117	(16.8)	96	(27.2)	
	Nx	126	(12.0)	81	(11.6)	45	(12.7)	<0.001
Life status (at 31/12/2016)	Alive	536	(51.0)	353	(50.6)	183	(51.8)	
	Dead	514	(49.0)	344	(49.4)	170	(48.2)	0.714

DCO: Death certificate only; TNM7: TNM Classification System 7th Edition.

**Table 2 ijerph-17-06697-t002:** Distribution of colorectal cancer patients according to diagnostic exams and preoperative tests.

		Sub-Site						
		Colorectal		Colon		Rectum		*p*-Value
		*n*	(%)	*n*	(%)	*n*	(%)	
**Total**		1050	(100.0)	697	(100.0)	353	(100.0)	
Colonoscopy	Not done	152	(14.5)	135	(19.4)	17	(4.8)	
	Done, complete	610	(58.2)	375	(53.8)	235	(66.8)	
	Done, incomplete	287	(27.4)	187	(26.8)	100	(28.4)	<0.001
Barium enema	Done	71	(6.9)	51	(7.5)	20	(5.8)	
	Not done	959	(93.1)	633	(92.5)	326	(94.2)	0.316
Computed Tomography Magnetic Resonance Imaging (CT MRI) Colonography	Done	146	(14.2)	27	(4.0)	119	(34.5)	
	Not done	882	(85.8)	656	(96.0)	226	(65.5)	<0.001
Number of lymph nodes examined	<11 lymph nodes	435	(42.7)	263	(38.7)	172	(50.7)	
	≤12 lymph nodes	583	(57.3)	416	(61.3)	167	(49.3)	<0.001
Lung imaging	Done	1025	(99.2)	679	(99.3)	346	(99.1)	
	Not done	8	(0.8)	5	(0.7)	3	(0.9)	0.698
Liver imaging	Done	1023	(98.9)	681	(99.4)	342	(98.0)	
	Not done	11	(1.1)	4	(0.6)	7	(2.0)	0.082
Brain imaging	Done	457	(44.4)	301	(44.2)	156	(44.8)	
	Not done	572	(55.6)	380	(55.8)	192	(55.2)	0.848
Skeleton imaging	Done	471	(45.8)	306	(44.9)	165	(47.4)	
	Not done	558	(54.2)	375	(55.1)	183	(52.6)	0.450
Pre-operative echography	Done	626	(60.8)	417	(61.1)	209	(60.2)	
	Not done	404	(39.2)	266	(38.9)	138	(39.8)	0.798
Pre-operative thoracic xRay	Done	858	(83.2)	576	(84.3)	282	(81.0)	
	Not done	173	(16.8)	107	(15.7)	66	(19.0)	0.180
Pre-operative thoracic CT	Done	840	(81.5)	538	(78.8)	302	(86.8)	
	Not done	191	(18.5)	145	(21.2)	46	(13.2)	0.002
Pre-operative abdominal CT	Done	940	(91.2)	623	(91.2)	317	(91.1)	
	Not done	91	(8.8)	60	(8.8)	31	(8.9)	0.974
Pre-operative MRI	Done	266	(26.1)	45	(6.6)	221	(64.4)	
	Not done	755	(73.9)	633	(93.4)	122	(35.6)	<0.001
Pre-operative echoendoscopy	Done	457	(44.4)	263	(38.5)	194	(56.1)	
	Not done	572	(55.6)	420	(61.5)	152	(43.9)	<0.001

**Table 3 ijerph-17-06697-t003:** Distribution of colorectal cancer patients according to treatment received.

		Sub-Site						
		Colorectal		Colon		Rectum		*p*-Value
		*n*	(%)	*n*	(%)	*n*	(%)	
**Total**		1050	(100.0)	697	(100.0)	353	(100.0)	
Surgery	Not done	173	(16.5)	106	(15.3)	67	(19.1)	
	Total colectomy	29	(2.8)	25	(3.6)	4	(1.1)	
	Hemi-colectomy	337	(32.2)	326	(46.9)	11	(3.1)	
	Anterior resection	179	(17.1)	16	(2.3)	163	(46.4)	
	Segmental resection	203	(19.4)	182	(26.2)	21	(6.0)	
	Abdomino-perineal resection	65	(6.2)	6	(0.9)	59	(16.8)	
	Other or unknown type	60	(5.7)	34	(4.9)	26	(7.4)	<0.001
Type of hospital admission	Planned	689	(79.2)	431	(73.3)	258	(91.5)	
	Emergency	181	(20.8)	157	(26.7)	24	(8.5)	<0.001
Mode of surgery	Open surgery	688	(79.4)	468	(79.7)	220	(78.9)	
	Laparoscopic surgery	178	(20.6)	119	(20.3)	59	(21.1)	0.766
Reasons for no surgery	Medical contraindications	17	(10.1)	11	(10.7)	6	(9.2)	
	Patient refusal	23	(13.7)	13	(12.6)	10	(15.4)	
	Advanced cancer	97	(57.7)	64	(62.1)	33	(50.8)	
	Other	23	(13.7)	11	(10.7)	12	(18.5)	
	No indications	8	(4.8)	4	(3.9)	4	(6.2)	0.545
Involvement of surgical margins	R0 resection	695	(97.7)	463	(97.9)	232	(97.5)	
	R1 resection	16	(2.3)	10	(2.1)	6	(2.5)	0.730
Resection of metastasis	R0 resection	64	(43.0)	48	(44.4)	16	(39.0)	
	R2 resection	54	(36.2)	41	(38.0)	13	(31.7)	
	R2: no resection	31	(20.8)	19	(17.6)	12	(29.3)	0.291
Colostomy	Done	246	(26.1)	89	(13.9)	157	(52.2)	
	Not done	697	(73.9)	553	(86.1)	144	(47.8)	<0.001
Type of colostomy	Permanent	129	(53.5)	47	(54.0)	82	(53.2)	
	Temporary	97	(40.2)	29	(33.3)	68	(44.2)	
	Alone. without resection	15	(6.2)	11	(12.6)	4	(2.6)	0.005
Chemotherapy	Done	485	(47.0)	280	(40.9)	205	(59.2)	
	Not done	546	(53.0)	405	(59.1)	141	(40.8)	<0.001
Modality of chemotherapy	Neo-adjuvant	126	(26.0)	14	(5.0)	112	(54.6)	
	Adjuvant	264	(54.4)	202	(72.1)	62	(30.2)	
	Perioperative	0	(0.0)	0	(0.0)	0	(0.0)	
	Palliative	95	(19.6)	64	(22.9)	31	(15.1)	<0.001
Reasons for no chemotherapy	Medical contraindications	104	(19.3)	76	(19.0)	28	(20.1)	
	Patient refusal	33	(6.1)	21	(5.3)	12	(8.6)	
	Other	89	(16.5)	67	(16.8)	22	(15.8)	
	No indications	313	(58.1)	236	(59.0)	77	(55.4)	0.516
Radiotherapy	Done	184	(17.8)	4	(0.6)	180	(52.2)	
	Not done	850	(82.2)	685	(99.4)	165	(47.8)	<0.001
Modality of radiotherapy	Neo-adjuvant	125	(67.9)	1	(25.0)	124	(68.9)	
	Adjuvant	46	(25.0)	0	(0.0)	46	(25.6)	
	Palliative	13	(7.1)	3	(75.0)	10	(5.6)	<0.001
Reasons for no radiotherapy	Medical contraindications	23	(2.7)	8	(1.2)	15	(9.1)	
	Patient refusal	12	(1.4)	3	(.4)	9	(5.5)	
	Other	65	(7.7)	43	(6.3)	22	(13.4)	
	No indications	744	(88.2)	626	(92.1)	118	(72.0)	<0.001
Targeted Treatment (TT)	Done	62	(6.0)	45	(6.6)	17	(5.0)	
	Not done	966	(94.0)	640	(93.4)	326	(95.0)	0.306

R0: No residual tumor; R1: Microscopic residual tumor; R2: Macroscopic residual tumor.

**Table 4 ijerph-17-06697-t004:** Number (and percentage) of colorectal cancer patients as a function of adherence to the quality indicators (QIs).

Quality Indicator	Sub-Site											
	Total				Colon				Rectum			
	No		Yes		No		Yes		No		Yes	
	*n*	(%)	*n*	(%)	*n*	(%)	*n*	(%)	*n*	(%)	*n*	(%)
QI1: the patient, following the CRC diagnosis, was assessed by the specific tumor commission before starting treatment.	85	(8.1)	961	(91.9)	63	(9.1)	632	(90.9)	22	(6.3)	329	(93.7)
QI2: complementary imaging tests were requested: colonoscopy, CT of the chest, abdomen and pelvis, and MRI of the pelvis.	399	(38.0)	651	(62.0)	258	(37.0)	439	(63.0)	141	(39.9)	212	(60.1)
QI3: the patient underwent surgery sooner than 30 days after the histological diagnosis.	280	(38.5)	448	(61.5)	201	(35.2)	370	(64.8)	79	(50.3)	78	(49.7)
QI4: the patient initiated neoadjuvant therapy (radiotherapy/chemotherapy or chemotherapy or radiotherapy) sooner than 30 days after the histological diagnosis.	95	(65.1)	51	(34.9)	8	(53.3)	7	(46.7)	87	(66.4)	44	(33.6)
QI5: the patient started adjuvant treatment sooner than 8 weeks after surgical treatment.	95	(36.1)	168	(63.9)	59	(29.2)	143	(70.8)	36	(59.0)	25	(41.0)
QI6: the patient underwent excision and analysis of at least 12 lymph nodes to allow for appropriate lymph node staging.	198	(25.4)	583	(74.6)	119	(22.2)	416	(77.8)	79	(32.1)	167	(67.9)
QI7: the patient, if diagnosed with a stage III colon carcinoma, underwent chemotherapy treatment.	56	(33.3)	112	(66.7)	56	(33.3)	112	(66.7)	-			
QI8: the patient, if diagnosed with stage II or III carcinoma of the rectum underwent radiotherapy/chemotherapy or radiotherapy treatment with neoadjuvant or adjuvant intent.	62	(26.6)	171	(73.4)	-				62	(26.6)	171	(73.4)
QI9: perioperative mortality, defined as patient death in the first 30 days after surgical treatment	821	(94.2)	51	(5.8)	550	(93.4)	39	(6.6)	271	(95.8)	12	(4.2)

**Table 5 ijerph-17-06697-t005:** Observed (OS) and net survival (NS) at 1, 3 and 5 years since diagnosis of colorectal cancer and relative excess risk of death (RER) as a function of adherence to the quality indicators (QIs).

Quality Indicator		Years Since Diagnosis									Relative Excess Risk of Death		
		1 Year			3 Years			5 Years			RER	95% CI	*p*-Value
		OS	NS	95% CI	OS	NS	95% CI	OS	NS	95% CI			
QI1	No	69.4	71.9	(60–80.8)	57.6	63.1	(49.7–73.9)	51.8	60.7	(45.9–72.6)	1		-
	Yes	77.4	79.8	(76.9–82.4)	60.1	66.2	(62.6–69.5)	52.0	62.4	(58.3–66.1)	1.21	(0.70–2.08)	0.495
QI2	No	66.3	68.8	(63.7–73.4)	49.8	55.9	(50.1–61.3)	43.3	54.6	(48.1–60.7)	1		-
	Yes	83.3	85.6	(82.3–88.3)	66.2	72.1	(67.9–75.9)	57.3	67.0	(62.1–71.3)	0.58	(0.46–0.73)	<0.001
QI3	No	88.6	91.5	(86.6–94.6)	75.7	83.9	(77.2–88.8)	66.8	81.1	(73–87)	1		-
	Yes	82.2	84.7	(80.6–87.9)	66.3	73.2	(67.9–77.7)	58.4	71.0	(64.9–76.2)	1.77	(1.17–2.66)	0.007
QI4	No	90.5	92.4	(83.4–96.6)	74.7	79.8	(68.4–87.5)	61.1	67.7	(54.9–77.6)	1		-
	Yes	90.2	91.5	(78.1–96.8)	68.6	72.4	(56.4–83.3)	58.8	63.2	(46.2–76.2)	1.27	(0.65–2.50)	0.479
QI5	No	96.8	99.3	(41–100)	76.8	83.2	(71.3–90.5)	66.3	77.3	(63.7–86.3)	1		-
	Yes	93.5	95.1	(89.5–97.8)	76.8	81.1	(73.2–87)	66.1	73.0	(63.9–80.1)	1.10	(0.61–1.98)	0.749
QI6	No	83.4	86.0	(79.6–90.5)	68.8	76.1	(67.9–82.4)	60.8	74.4	(64.7–81.9)	1		-
	Yes	88.4	91.0	(87.9–93.4)	72.1	79.5	(75.1–83.2)	62.1	74.8	(69.6–79.3)	1.00	(0.68–1.47)	0.995
QI7	No	69.6	75.3	(59.3–85.7)	46.4	58.7	(39.7–73.5)	42.9	64.7	(40.2–81.3)	1		-
	Yes	95.5	97.4	(88.6–99.4)	77.7	82.8	(72.6–89.4)	70.5	78.9	(67.2–86.8)	0.33	(0.16–0.70)	0.004
QI8	No	79.0	82.7	(68.7–90.8)	69.4	81.2	(62.5–91.1)	58.1	84.0	(56.1–94.9)	1		-
	Yes	93.5	95.7	(88.2–98.5)	77.4	83.4	(73.5–89.9)	66.9	75.8	(64.3–84.1)	0.92	(0.40–2.14)	0.848
Overall adherence (≥75% of indicators)	No	71.1	73.8	(69.2–77.7)	54.1	60.2	(54.9–65.2)	47.0	57.7	(51.7–63.2)	1		-
	Yes	81.4	83.5	(79.9–86.5)	64.7	70.6	(66–74.6)	56.0	65.9	(60.8–70.6)	0.35	(0.28–0.45)	<0.001

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
