# Peer review of "Adherence to Clinical Practice Guidelines and Colorectal Cancer Survival: A Retrospective High-Resolution Population-Based Study in Spain"

_ijerph, 2020, doi:10.3390/ijerph17186697_

Round 1

Reviewer 1 Report

The article is very relevant given the frequency of this type of cancer in the world and in Spain. The importance of population health and behavioral health in early detection and prevention of this condition has been demonstrated in most healthcare systems around the world. The study shows the significant advantage of a diagnostic in an early stage of the disease.

The development of clinical guidelines for this condition are well known and established and the adherence to those guidelines represent a cost-effective way to reduce the burden of this disease. The research methods, the findings, and discussion are well designed, executed and reported in the article. In the discussion section there are clear recommendations based on the results of this study that should help healthcare providers and health systems policy officers improve the early detection and treatment of this condition following the specific clinical guidelines mentioned as significant in the study.

The dissemination of this study should help the authors and other researches interested in this topic, to develop further research initiatives to address the limitations indicated in the study by expanding it to all the regions of Spain progressively, starting with those regions or communities that have a higher incidence of this condition. Also, to compare the results and specific practices between the different regions and benchmark with international experiences as well. The comparison with other related health conditions could also be an interesting study in the future to learn from other similar possible strategies and the collective impact that they have in population health.  

Author Response

We thank the reviewer for the positive feedback on our manuscript and for recognizing the potential value of the results. We have added the mentioned future research initiatives to the discussion on p. 14.

“In addition, comparing the results across provinces was not among the goals of the current study but it should be addressed in future research because regional differences in survival have been observed for other cancers [34]. It would be especially relevant to do this for the regions with highest incidence and mortality from colorectal cancer and further benchmark with results from other countries. There is important variability in treatments administered both within and between countries, as is the case for radiotherapy [35–37] and chemotherapy [15].

Reviewer 2 Report

This is a population-based study which analyze the degree of adherence to clinical practice guidelines for colorectal cancer (CRC) and investigate its relationship with survival in all incident CRC cases diagnosed in 2011 in two provinces in Spain.

The manuscript is well written and the results are interesting and clearly presented. This is a retrospective high-resolution study conducted with 1050 incident CRC cases from the cancer registries of Granada and Girona based to nine quality indicators of the relevant CRC guidelines and with a follow-up of 5 years.

Similar surveys have been published relevant to the adherence to clinical practice guidelines for colorectal cancer in a Canadian Province and in the Netherlands. The authors should make a reference in the manuscript text about these studies indicating which are the differences and the added value for their survey.

Minor comments: 

  • Tables 4 and 5 need to be formated since some brackets are shifted.

Author Response

Review comments:

This is a population-based study which analyze the degree of adherence to clinical practice guidelines for colorectal cancer (CRC) and investigate its relationship with survival in all incident CRC cases diagnosed in 2011 in two provinces in Spain.

The manuscript is well written and the results are interesting and clearly presented. This is a retrospective high-resolution study conducted with 1050 incident CRC cases from the cancer registries of Granada and Girona based to nine quality indicators of the relevant CRC guidelines and with a follow-up of 5 years.

Answer: Thank you for the positive feedback on our manuscript.

Similar surveys have been published relevant to the adherence to clinical practice guidelines for colorectal cancer in a Canadian Province and in the Netherlands. The authors should make a reference in the manuscript text about these studies indicating which are the differences and the added value for their survey.

Answer: Thank you for this constructive suggestion, we have added this information on page 11:

“Adherence to clinical practice guidelines for CRC has recently been examined in other countries including Canada [14] and the Netherlands [15]. In particular, a population-based study in a Canadian province examined adherence to adjuvant chemotherapy in patients with stage II or III colorectal cancer [14], whereas in the Netherlands a survey of medical oncologists examined adherence to clinical guidelines for systemic treatment for high risk stage II and III colon and metastatic colorectal cancer [15]. However, these studies did not investigate the relationship between adherence and survival. Hence, the current study adds valuable information regarding the implications of a broad set of clinical guidelines for patient survival, using population-based data and including all patients in the selected regions, regardless of stage at diagnosis.”

We have also mentioned or compared the obtained results in these studies to ours on p. 13:

“However, in our study only 66.7% of patients underwent adjuvant treatment, percentage that is still greater than that reported by other European [4] and North-American [14] cancer registries.”

..and p. 14:

“There is important variability in treatments administered both within and between countries, as is the case for radiotherapy [35–37] and chemotherapy [15].”

Minor comments:

Tables 4 and 5 need to be formated since some brackets are shifted.

Answer: We have re-formatted the tables 4 and 5 (pages 25 and 26) and hope that everything is now in place.

Reviewer 3 Report

Colorectal cancer is a common cancer worldwide, and it is considered a very important issue as a study that evaluated guidelines for treatment that can increase survival rates that can reduce mortality. I am honored to evaluate this important study.

I Attached a simple revision as a file.

Author Response

Reviewer comments:

  1. Title - I hope that the title will be more detailed, and the research method and content will be briefly understood.

Answer: Thank you for this suggestion, we have modified the title to: “Adherence to clinical practice guidelines and colorectal cancer survival: A retrospective high-resolution population-based study in Spain”

  1. Abstract - At the end of the abstract, the meaning and suggestions for this research result should be added.

Answer: The two key implications of the results are now mentioned at the end of the abstract (p. 3): “Ordering complementary imagining tests that improve staging and treatment choice for all CRC patients and adjuvant chemotherapy for stage III colon cancer patients could be especially important. In contrast, controlled delays in starting some treatments appear not to decrease survival.”

We have also introduced some other minor editions to the abstract to be able to conform to the maximum number of words allowed by the journal.

  1. Introduction - It is necessary to add an introduction title according to the form of this journal. - “Adherence to clinical practice guidelines and colorectal cancer survival” The title needs to be deleted from the introduction

Answer: Thank you, we have now deleted the title and headed the section “Introduction” as per journal standards (p. 4).

  1. Materials and Method - Please explain why the subject was 15 years of age or older, especially if there are special reasons for choosing Granada and Girona in Spain.

Answer: We have added this information to page 5.

Limit set at 15 years: The method for calculating net survival is designed from a theoretical point of view to work with the age group 15-99 years to avoid the distortion that can be produced by the generally very (in the case of those under 15 years) and very high (over 99 years) mortality rates. Anyway, all the incident cases considered for the study were included within these age limits (range 15-97 years).

Granada and Girona were selected among the seven Spanish cancer registries participating in the European High Resolution studies to represent Southern (Granada) and Northern (Girona) Spain and because they have similar population sizes, thereby contributing a similar number of CRC cases to the analysis.

  1. Data collection & Data analysis - To help readers understand the statistical analysis method, it should be organized in a simpler yet more explanatory way.

Answer: We have introduced some additional explanation to the section on statistical Analyses and we hope it is now more understandable (see changes marked on p. 7-8).

  1. Result - Each table needs to be rearranged for better readability.

Answer: It appears that in the previous version some of the table formatting had moved on submission, decreasing the readability of the tables. We hope that this problem has now been fixed.  If the reviewer refers to some other issues with the tables, please let us know, so that we can edit accordingly.

  1. Conclusion The conclusions need to be explained in more detail and include suggestions.

Answer: We have now expanded the summary of conclusions at the start of the discussion section to add more detail and also mention the implications of the results for patient survival (see marked changes on p. 10).

Results showed that overall adherence to the clinical practice guidelines (on ≥75% of indicators) improved survival, reducing excess risk of death by 60%-65% (depending on whether all patients are considered or only those diagnosed in stages II and III, respectively).  Detailed analyses of the separate indicators suggested that ordering complementary imagining tests that improve staging and treatment choice for all CRC patients and adjuvant chemotherapy for stage III colon cancer patients could improve survival. In contrast, controlled delays in starting some treatments appeared not to decrease survival.”

Suggestions for future research have been added to the discussion as well on p. 14:

“In addition, comparing the results across provinces was not among the goals of the current study but it should be addressed in future research because regional differences in survival have been observed for other cancers [34]. It would be especially relevant to do this for the regions with highest incidence and mortality from colorectal cancer and further benchmark with results from other countries. There is important variability in treatments administered both within and between countries, as is the case for radiotherapy [35–37] and chemotherapy [15].”

Reviewer 4 Report

Authors of the article presented a population-based study in Spain. Their main result is that "Overall adherence to the guidelines significantly reduced the excess risk of death".

The manuscript is well written, a few typos should be corrected, and some minor things should have to be addressed:

  • Results: p-values should be presented in the text as well.
  • While some of them are basically textbook results, but within the descriptive sections (Tables 1-3) p-values should be also presented both in tables and text, where authors describe any difference between groups.
  • Line 146: Elandt-Johnson is the correct name of the method, not Edlant-Johnson
  • Table 4: what does „Si” mean (I assume "Yes")
  • line 228: [4] [15,16] should be [4,15,16]
  • line 236-237: „The 62% of patients for whom the complementary imaging studies recommended were requested (QI2) had better survival...”  -> sentence should be revised

Author Response

Reviewer comments:

Authors of the article presented a population-based study in Spain. Their main result is that "Overall adherence to the guidelines significantly reduced the excess risk of death". The manuscript is well written, a few typos should be corrected, and some minor things should have to be addressed.

Answer: We thank the reviewer for the positive feedback on our manuscript.

Results: p-values should be presented in the text as well. While some of them are basically textbook results, but within the descriptive sections (Tables 1-3) p-values should be also presented both in tables and text, where authors describe any difference between groups.

Answer: We have now added p-values in the text whenever we refer to statistical comparisons (see changes markedon pages 9-10). We have also added p-values for the comparisons between subsites to tables 1 to 3 as requested (however, please note that these comparisons are not commented in the text but are available in the tables for interested readers).

Line 146: Elandt-Johnson is the correct name of the method, not Edlant-Johnson

Answer: This typo has been corrected, thank you for noticing.

Table 4: what does „Si” mean (I assume "Yes")

Answer: Indeed, “Si” has been changed to “Yes” in Table 4, apologies.

line 228: [4] [15,16] should be [4,15,16]

Answer: This typo has been corrected.

line 236-237: „The 62% of patients for whom the complementary imaging studies recommended were requested (QI2) had better survival...”  -> sentence should be revised

Answer: We have now revised the sentencesplitting it into two for better readability (p. 11): “Complementary imaging studies (colonoscopy, CT of the chest, abdomen and pelvis, and MRI of the pelvis) were requested for 62% of patients (QI2). These patients had better survival, in particular a 42% reduced excess risk of dying, compared to patients for whom no complementary imaging studies were requested.”
